# The Oil Price and Trade Nexus in the Gulf Co-Operation Council Countries

**Tarek Tawfik Yousef Alkhateeb [1,2] and Haider Mahmood [1,*]** 

[1]   College of Business Administration, Prince Sattam bin Abdulaziz University, 173,
     Alkharj 11942, Saudi Arabia; tkhteb1@gmail.com
[2]   Department of Agriculture Economics, Kafr Elshiekh University, Kafr Elshiekh 33511, Egypt
*    Correspondence: h.farooqi@psau.edu.sa

**Abstract:** The present study investigates the asymmetrical impacts of oil prices and real exchange rates on the trade balances in the Gulf Co-operation Council countries. Using panel estimates, we found the symmetrical positive effect from the oil prices and the asymmetrical positive effect from the real exchange rates on the trade balances. For country-specific results, increasing oil prices showed a positive effect on the trade balances in Oman, Saudi Arabia, and the UAE and a negative effect in Kuwait. Decreasing oil prices carried a positive relationship with the trade balances in Bahrain, Oman, Qatar, and the UAE. The oil prices showed an asymmetrical impact on the trade balances in all countries, except Saudi Arabia. Moreover, the depreciation helped to improve the trade balances in Bahrain, Oman, Qatar, and the UAE. The appreciation worsened the trade balances in Oman, the UAE, and Saudi Arabia and improved the trade balance in Kuwait. Moreover, the asymmetrical relationships between the real exchange rates and the trade balances were corroborated in all of the investigated countries.

**Keywords:** trade balances; real exchange rates; oil prices; asymmetry

## 1. Introduction

In a seminal paper, Svensson [1] argued that trade balances in oil-importing countries might be worsened by the temporary rise in oil prices. However, this effect is uncertain in the case of the perpetual increase in the prices of oil. Oil price changes might re-distribute the income between oil exporting and importing economies [2]. Timilsina [3] established that increasing oil prices harmed global income and accelerated the exports of oil-intensive economies. Bodenstein et al. [4] probed the influence of oil prices on the trade balances and corroborated that increasing oil prices benefited the oil exporters' income because of the wealth-transfer effect. In this way, it reduced the welfare of oil importers by a reduction in consumption. This process might also depreciate the oil importers' real exchange rates and would deteriorate the oil–trade balances. In emerging economies, importers usually put their currency reserves in oil-exporting economies to stabilize the foreign value of their money. Increasing oil prices could develop expectations about the appreciation in the value of the reserves in oil-exporting economies [5]. Therefore, oil prices may change the balance of payments through changing trade, the assets' values, and capital flows [6].

The price of oil is a major influencing factor in oil-exporting economies, which rely on oil revenue to support their income, government spending, and exports. Increasing oil prices could boost the value of oil exports, as oil demand is mostly price inelastic, as oil is a major source of industrial energy usage [7] and may have a positive impact on the trade balance. However, increasing oil prices also support the income of oil-exporting countries which could raise import and inflation rates [8]. Resultantly, it could deteriorate the trade balance. Korhonen and Ledyaeva [8] have validated the positive effect of increasing oil prices on imports of oil-producing nations. Moreover, increasing oil prices may lead to the

appreciation of oil exporters' currencies. This appreciation would increase the price of products from oil-producing countries in the world market and could reduce the demand for their exports. Collier [9] elaborated that natural resources were one reason for the slow growth of resource-abundant economies. In addition, Le and Chang [6] argued that increasing oil prices would slow down the economic growth of oil-importing economies, which would consequently reduce demand for exports from oil-exporting economies. Increasing oil prices may have a positive or adverse effect on the trade balance, and the net effect of increasing oil prices is an empirical question for any oil-exporting country.

Theoretical arguments favor the positive effect of increasing oil prices on the trade balances in oil-exporting countries [3–5,7] and also discuss the negative role of increasing oil prices on the trade balances [6,8]. In another argument, the decreasing price of oil does not necessarily have with certainty the same sign or magnitude of effect compared to the effect of increasing oil price. Thus, asymmetry is expected, and the literature has also documented an asymmetrical impact of oil prices on the trade balance [7,10–14]. Rafiq et al. [7] claimed that decreasing oil prices would support the trade balances of oil-exporting countries if a percentage of the increased demand was greater than the percentage of the decline in oil prices. Therefore, the different signs and/or magnitudes of coefficients of increasing and decreasing oil prices are expected on oil exporters' trade balances. Rafiq et al. [7] argued that oil price fluctuations could have a favorable impact on oil exporting countries, and stable oil prices might have a positive effect on the oil-importing countries. Using empirical evidence, Baek and Kwon [12] corroborated that the symmetrical impact of oil prices on the trade balance was found to be insignificant and was significant in the asymmetrical analyses.

In the relationship between oil prices and macroeconomic performance, the bulk of the literature has investigated the relationship between oil prices and the stock market in the Gulf Co-operation Council (GCC) countries [15–25]. Moreover, the literature also probed the effect of oil prices on economic growth [26], personal consumption [27], foreign direct investment inflows [28], government spending and fiscal deficits [29,30], employment [31], military spending [32], external surpluses [33], exchange rates [25], energy consumption [34], and on energy depletion [35]. Gazdar [36] probed the effect of the oil term of trade shocks on economic growth, and Jouini [37] explored the effect of trade openness on the economic growth of the GCC region. Moreover, Aloui [38] corroborated the bidirectional relationship between exchange rates and income as well as exchange rates and inflation in Saudi Arabia.

The testing of the effect of oil prices on the trade balance is limited in the global literature [5–7,11–14,39]. Salisu [40] investigated the asymmetrical impact of oil prices on inflation in oil-exporting countries, and Hatemi-J and El-Khatib [41] tested the asymmetrical impact of oil prices on the exchange rates. However, testing of the asymmetrical impact of oil prices on the trade balance is relatively scant [7,11–14]. Some studies have probed an asymmetrical impact of oil prices on the trade balances in oil-exporting economies [7,11,13]; however, no study focused on the GCC countries' panel in particular. Thus, we were highly motivated to find the asymmetrical effects of oil prices and real exchange rates on the trade balances in a panel of the GCC countries and country-specific estimates as well. Figure 1 shows the hypothesized model that the Dutch Disease phenomenon and oil prices would affect the trade balance. Moreover, the oil prices and real exchange rates can be converted into increasing and decreasing series to test their asymmetrical effects on the trade balances in the GCC countries.

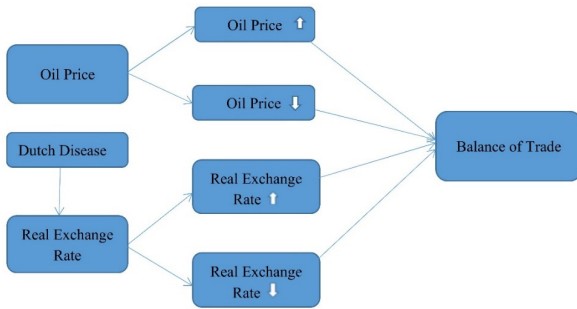

**Figure 1.** Hypothesized model.

## 2. Literature Review

A study by Kilian et al. [5] might be considered a pioneering piece of empirical research to verify the impact of oil prices on the external balances. The price of oil was found as a major factor to determine the trade balances of oil exporting and importing economies. Rafiq et al. [7] instigated the testing of the asymmetrical impact of oil prices on the trade balances in 28 oil-exporting and 40 oil-importing countries. Increasing oil prices improved the oil–trade balances and harmed the non-oil and total trade balances of oil-exporting countries. In contrast, decreasing oil prices was found to be helpful in improving the oil and total trade balances of oil-exporting economies. It means that a percentage of increased demand for oil exports was found to be greater than a percentage fall in oil prices. In the case of oil importers, decreasing oil prices harmed the oil and total trade balances.

Baek et al. [13] probed the asymmetrical impact of oil prices on the oil, non-oil, and total trade balances in some oil-exporting economies. In the case of Iran, increasing (decreasing) oil prices had a positive (a negative) impact on the oil–trade balance. In Saudi Arabia, oil prices had a symmetrical negative effect on the oil and non-oil trade balances. To highlight the importance of asymmetrical analyses, Baek and Kwon [12] examined the impact of oil prices on the oil, non-oil, and total trade balances in some African oil-exporting countries. They could not find the impact of oil prices on the trade balances in the symmetrical analysis. In the nonlinear analyses, they found the asymmetrical impact of oil prices on the oil–trade balances in most of the investigated countries. However, these effects were mostly found as insignificant for the non-oil and total trade balances. Jibril et al. [11] probed the asymmetrical effects of oil demand, oil supply, and aggregate demand shocks on the trade balances. In the analyses of oil exporters, they established that increasing oil supply shock harmed the oil and total trade balances, and decreasing oil supply shocks had positive effects on the non-oil and total trade balances. Increasing aggregate demand showed a positive effect on the oil and total trade balances. Positive oil demand shocks had positive effects on the oil, non-oil, and total trade balances, and negative shocks had negative effects on all types of trade balances. The opposite effects were found in the case of oil importers.

Le and Chang [6] investigated the causality between oil prices and the trade balances in Japan, Singapore, and Malaysia. In the causality analyses, the oil prices caused the oil–trade balances in all investigated countries, caused the total trade balance in Malaysia, and caused the non-oil trade balance in Japan. Ahad and Anwer [14] studied the non-linear effects of increasing and decreasing industrial production index, inflation, and oil prices on the trade balance in Pakistan. They found the positive and asymmetrical impacts of oil prices and inflation on the trade balance with the different magnitude of coefficients. The negative asymmetrical effect of the industrial production index was also corroborated on the trade balance in Pakistan. Nanovsky [39] examined the oil prices in the gravity equation of the world trade and found that increasing oil prices significantly contributed to the shipping and transport costs of international trade. Therefore, increasing oil prices restricted trade among nearby countries, and the decreasing oil prices expanded the global trade.

In the GCC region, Metwally [33] probed a relationship between oil price downturns and external surplus and found that the fall in oil exports significantly decreased the external surpluses. Al Rasasi [42] tested a nonlinear relationship between oil prices and real exchange rates and found that oil prices appreciated the local currency. Metwally and Perera [29] explored the impact of decreasing oil prices on government spending. They found that during the oil price declines, governments of the GCC countries increased the spending to maintain the economic growth rates. El Mahmah and Kandil [30] argued that the GCC countries managed the fiscal deficit by issuing the debt. They found that lag of primary balances, debts, and oil prices showed positive effects on the primary balances. Hence, the issuance of debt helped to adjust the primary balances and increasing oil prices also improved the primary balances in the GCC countries.

Arouri et al. [16] investigated the effect of oil prices on the Stock Market Indices (SMI) and found the nonlinear relationship in Saudi Arabia, the UAE, Qatar, and Oman. This relationship was also switching because of the oil price movements. Arouri and Rault [17] probed and corroborated

a long-run relationship between oil prices and SMI in all GCC countries except Saudi Arabia. Arouri et al. [18] re-investigated this relationship using a period 2005–2010. They found a significant relationship between oil prices and Stock Market Returns (SMRs). Fayyad and Daly [19] investigated the relationship between oil price shocks and SMRs in the UK, the US, and the GCC region during 2005–2010. They found that the predictability of SMRs improved with the rising oil prices and most of the countries' SMRs showed responsiveness to oil price shocks. Louis and Balli [20] probed a relationship between oil prices and SMI volatility in the GCC market and found a mild-to-strong adjustment between the variables. Akoum et al. [13] explored and found a strong relationship between the oil returns and SMRs in the GCC region.

Nusair and Al-Khasawneh [22] found the asymmetry in the relationship between oil prices and SMRs. Moreover, this relationship was also switching over the investigated distribution. A positive relationship was validated between increasing oil prices and SMRs in the higher quantiles and between decreasing oil prices and SMRs in the lower quantiles. Before and during the oil prices' slump of 2014, Siddiqui et al. [23] found the asymmetrical relationships between the oil prices and SMI in the GCC countries. The oil prices had a positive (a negative) effect on the SMI in oil-exporting (oil-importing) countries. During the oil prices crisis period, they found that decreasing oil prices had a larger effect on the oil exporters and increasing oil prices had a larger impact on oil importers. During 2010–2017, Mokni and Youssef [21] examined the relationship between oil prices and the SMI in the GCC region and found that the interdependence of oil prices and SMI was increased after the 2014-oil price shocks.

Nusair [26] found the positive and inelastic effects of increasing and decreasing oil prices on the economic growth in Kuwait and Qatar. Mahmood and Zamil [27] explored the impacts of oil prices and oil price slumps of 2008 and 2014 on personal consumption in Saudi Arabia. They found the positive impact of oil prices and the insignificant effect of oil slumps on personal consumption. It was also concluded that increasing (decreasing) oil prices would increase (decrease) personal consumption. De et al. [43] explored the relationship among oil prices, remittances, and income in the GCC countries. They found that non-oil income had a positive effect on the remittances' outflow and oil income had a positive effect on the non-oil income. They also found the positive co-movements between oil prices and remittances, and the volatility of oil prices was found to be more than the volatility of remittances. In disaggregated analyses, they found that the construction sector and public sector played a great role in increasing remittances' outflows. Alkhateeb and Mahmood [35] probed the oil prices and energy depletion nexus and found that oil prices had a positive and asymmetrical effect on the energy depletion in the GCC region. Erdogan et al. [32] probed the impact of oil price volatility on military expenditures in Bahrain, Kuwait, KSA, and Oman. They found the positive effect of oil price volatility on the military expenditures in Kuwait, KSA, and Oman and the decreasing oil prices had a negative effect on military expenditures. Moreover, economic growth, trade openness, fixed capital formation, and military expenditure in Middle East and North Africa (MENA) countries had a mix of positive and negative effects on the military expenditures in the investigated GCC countries.

Mahmood and Alkhateeb [28] found a positive impact of oil prices on the foreign direct investment inflows of Saudi Arabia. Further, the financial market supported the foreign direct investment inflows and domestic investment harmed it. Alkhateeb et al. [31] studied the asymmetrical effect of oil prices on the employment generation in Saudi Arabia and found that both increasing and decreasing oil prices had a positive and inelastic effect on employment in the Kingdom. Moreover, asymmetry was corroborated with the different magnitude of coefficients. Comparatively, increasing oil prices had a greater impact on employment than that of decreasing oil prices. Further, economic growth also supported employment in the Kingdom. Table 1 shows a summary of literature review.

**Table 1.** Literature summary

| Authors | Countries | Major Findings |
|---------|-----------|----------------|
| *The literature on testing the symmetrical relationship between oil prices and trade* | | |
| Kilian et al. [5] | 26 oil-exporting and 12 oil-importing countries | Oil demand and supply shocks were majorly responsible for a change in the external balances of both oil exporting and importing economies. |
| Le and Chang [6] | Japan, Singapore, and Malaysia | Oil prices caused oil trade balances in all investigated countries. |
| Nanovsky [39] | 63 industrialized countries | Increasing oil prices restricted trade in nearby countries and vice versa. |
| *The literature on testing the asymmetrical relationship between oil prices and trade* | | |
| Rafiq et al. [7] | 28 oil-exporting and 40 oil-importing countries | The asymmetrical effect of oil prices was found in both oil-exporting and importing countries. |
| Baek et al. [13] | Organization of the Petroleum Exporting Countries (OPEC) members | The asymmetrical effect of oil prices on oil trade was found in some of the investigated countries. |
| Baek and Kwon [12] | African oil-exporter countries | The asymmetrical effect of oil prices on oil trade was found. |
| Jibril et al. [11] | 25 oil-exporting and 76 oil-importing countries | Increasing (decreasing) oil demand and supply shocks had negative (positive) effects in oil-exporting and vice versa in oil-importing countries. |
| Ahad and Anwer [14] | Pakistan | A positive and asymmetrical relationship was found between oil prices and the trade deficit. |
| *The literature on oil prices-related studies in GCC countries* | | |
| Metwally [33] | GCC countries | A positive relationship was found between oil exports and external surpluses. |
| Al Rasasi [42] | GCC countries | Increasing oil prices resulted in appreciation. |
| Metwally and Perera [29] | GCC countries | A negative relationship was found between oil prices and public spending. |
| El Mahmah and Kandil [30] | GCC countries | A positive relationship was corroborated between oil prices and fiscal primary balances. |
| Arouri et al. [16] | GCC countries | A nonlinear positive relationship was found between oil prices and SMI in 4 out of 6 GCC countries. |
| Arouri and Rault [17] | GCC countries | A positive relationship was found between oil prices and SMI in 5 out of 6 GCC countries. |
| Arouri et al. [18] | GCC countries | A positive relationship was found between oil prices and SMRs. |
| Fayyad and Daly [19] | UK, US, and GCC region | Oil prices improved the predictability of SMRs. |
| Louis and Balli [20] | GCC countries | A mild to a strong relationship was found between oil prices and SMI. |
| Akoum et al. [13] | GCC countries | A positive relationship was found between oil and SMRs. |
| Nusair and Al-Khasawneh [22] | GCC countries | A positive and asymmetrical relationship was found between oil prices and SMRs. |
| Siddiqui et al. [23] | GCC countries and the four largest Asian oil importers | A positive and asymmetrical relationship was found between oil prices and SMI in most of the GCC stock markets. |
| Mokni and Youssef [21] | GCC countries | A relationship between oil prices and SMRs became stronger after the 2014-oil price slump. |
| Nusair [26] | GCC countries | A positive and asymmetrical relationship was found between oil prices and economic growth. |
| Mahmood and Zamil [27] | Saudi Arabia | A positive relationship was found between oil prices and personal consumption. |
| De et al. [43] | GCC countries | A positive relationship was found between oil prices and remittances. |
| Alkhateeb and Mahmood [35] | Saudi Arabia | A positive and asymmetrical relationship was found between oil prices and energy depletion. |
| Erdogan et al. [32] | Bahrain, Kuwait, KSA, and Oman | A positive relationship was found between oil prices and military spending in Kuwait, KSA, and Oman. |
| Mahmood and Alkhateeb [28] | Saudi Arabia | A positive relationship was found between oil prices and foreign direct investment inflows. |
| Alkhateeb et al. [31] | Saudi Arabia | A positive relationship was found between oil prices and employment. |

The literature has shown the importance of oil prices in determining the trade balances of oil-exporting and oil-importing countries [5,6,39]. The literature also highlighted the importance of testing the asymmetry in the relationship between oil prices and trade [7,11–14]. Particularly, Baek and Kwon [12] corroborated the necessity of testing the asymmetrical effect of oil prices on trade in the case of oil-exporting countries. Some studies included the GCC countries in the panel of oil-exporting countries to investigate the asymmetrical relationship between oil prices and trade [7,11,13]. However, a focused study is still missing in the literature to test the asymmetrical relationship in a particular case of the GCC region. So, we explore the asymmetrical effect of oil prices on trade balances in the GCC

countries. Moreover, the asymmetrical effect of the real exchange rates is also included in the model of oil prices and trade to ensure a significant contribution to the GCC literature.

## 3. Methods

We followed the empirical model of Baek and Kwon [12] to explore the linkages among oil prices, the real exchange rates, and trade balances. Baek and Kwon [12] assumed the asymmetrical effect of oil prices and the symmetrical effect of the real exchange rates on the trade balances. Extending this model, we also incorporated the asymmetrical impact of the real exchange rates in the following way:

$$BOT_{it} = f(POP_{it}, NOP_{it}, PRER_{it}, NRER_{it}) \tag{1}$$

Balances of Trade ($BOT_{it}$) is a natural logarithm of the ratio of (exports/imports) of goods and services. This ratio represents the real trade balances and it is not affected by outliers [13]. The increasing ratio shows the improvement in the trade balances and the decreasing ratio means the deterioration in the trade balances. Further, OPEC Oil Prices ($OP_{it}$) are utilized in the model, and Positive Oil Prices ($POP_{it}$) and Negative Oil Prices ($NOP_{it}$) are the positive and negative variables of oil prices. The nominal exchange rate is defined as the units of currency of *i* GCC country to buy one US dollar and is converted into Real Exchange Rates ($RER_{it}$) by multiplying a ratio of consumer price indices (the base year of 2010) of the USA and *i* GCC country. In this definition, the positive movements of $RER_{it}$ mean a real depreciation of the local currency and the negative movements of $RER_{it}$ show a real appreciation. Positive Real Exchange Rates ($PRER_{it}$) and Negative Real Exchange Rates ($NRER_{it}$) are the positive and negative variables of $RER_{it}$. *i* presents six GCC countries. *t* denotes the period of maximum available data. We could collect the data for KSA, Kuwait, and Oman during 1970–2018, for the UAE during 2001–2018, for Bahrain during 1980–2018, and for Qatar during 1994–2018. The World Bank [44] and the Government of Saudi Arabia [45] are consulted to collect the data. Equation (1) can be estimated as a panel. Further, time-series analyses would be performed for each GCC country's model as well.

Before starting time and panel estimations, we test the nonlinearity or dependence of time series through the Broock–Dechert–Scheinkman (*BDS*) test proposed by Broock et al. [46]:

$$BDS_{e,m} = \sqrt{N}[COR_{e,m} - (COR_{e,1})^m] / \sqrt{V_{e,m}} \tag{2}$$

*COR* is a spatial correlation between time intervals with m-dimensional space with a difference $[COR_{e,m} - (COR_{e,1})^m]$, which is normally distributed with a variance $V_{e,m}$. Equation (2) would be tested with a null hypothesis ($H_0$): series has independent distribution. A rejection of $H_0$ would corroborate that series is nonlinear.

If the *BDS* test corroborated the non-linearity in series, then we would generate the increasing ($PX_t$) series by the partial sum of positive deviations in a mother series ($X_t$) and decreasing ($NX_t$) series by the partial sum of negative variations in a mother series ($X_t$), following Shin et al. [47]:

$$PX_t = \sum_{i=1}^{t} \Delta X_i^+ = \sum_{i=1}^{t} \max(\Delta X_i, 0) \tag{3}$$

$$NX_t = \sum_{i=1}^{t} \Delta X_i^- = \sum_{i=1}^{t} \min(\Delta X_i, 0) \tag{4}$$

After that, we test the issue of stationarity in the series. We apply the Ng and Perron [48] test which relies on 4 test statistics and may provide efficient results in the small sample size. The test-statistics are as follows:

$$MZ_a = [(Y_T^d/T) - f_0] / [2\sum_{t-2}^{T} (Y_T^d)^2 / T^2] \tag{5}$$

$$MSB = \sqrt{\sum_{t-2}^{T} (Y_T^d)^2 / T^2 * f_0} \tag{6}$$

$$MZ_t = MZ_a * MSB \tag{7}$$

$$MZT = [\bar{c}^2 \sum\nolimits_{t-2}^{T} (Y_T^d)^2 / T^2 + [(1-\bar{c})/T] * (Y_T^d)^2 / f_0 \tag{8}$$

The Equations (5)–(8) can be tested with $H_0$: Non-stationary series. A rejection of $H_0$ would corroborate a stationary series. After the confirmation of the order of integration in the model, we may develop the non-linear Autoregressive Distributive Lag (ARDL) for Equation (1) as follows:

$$\Delta BOT_t = \beta_0 + \beta_1 BOT_{t-1} + \beta_2 POP_{t-1} + \beta_3 NOP_{t-1} + \beta_4 PRER_{t-1} + \beta_5 NRER_{t-1}$$
$$+ \sum\nolimits_{j=1}^{n1} \gamma_{1j} \Delta BOT_{t-j} + \sum\nolimits_{j=0}^{n2} \gamma_{2j} \Delta POP_{t-j} + \sum\nolimits_{j=0}^{n3} \gamma_{3j} \Delta NOP_{t-j}^2 + \sum\nolimits_{j=0}^{n4} \gamma_{4j} PRER_{t-j} \tag{9}$$
$$+ \sum\nolimits_{j=0}^{n5} \gamma_{5j} \Delta NRER_{t-j} + \psi_t$$

Equation (9) can be regressed after allocation of optimum lag lengths to each differenced variable using the Schwarz Information Criterion (SIC) as per the methodology of Pesaran et al. [49]. Then, cointegration would be tested on the $H_0$: No-cointegration ($\beta_1 = \beta_2 = \beta_3 = \beta_4 = \beta_5 = 0$). A rejection of $H_0$ might corroborate the cointegration and long-run impacts can be estimated by normalizing the $\beta_k$ by $\beta_1$. Then, the short-run effects can also be assessed through estimated $\gamma$.

Before time series analyses, our objective is to test the hypothesized relationship in the whole panel of GCC countries. In the panel estimations, we apply the panel unit root Levin–Lin–Chu (LLC) test of Levin et al. [50]:

$$\Delta y_{it} = \alpha + \varphi y_{it-1} + \sum\nolimits_{L=1}^{pi} \phi_{iL} \Delta y_{it-L} + \iota_{mi} d_{mt} + e_{it} \tag{10}$$

Here, $y_{it}$ is a panel series with $i$ cross-sections and $t$ time dimensions. $p_i$ is the lag length which could vary across the countries and $d_{mt}$ is a deterministic variable. Equation (10) would be tested by the adjusted *t*-statistics with $H_0$: all cross-sections have the unit root. A rejection of $H_0$ would corroborate the stationary of the panel series. After that, the results of the LLC test can be verified with the Im–Pesaran–Shin (IPS) test of Im et al. [51] as follows:

$$\Delta y_{it} = \eta + o y_{it-1} + \sum\nolimits_{j=1}^{pi} \phi_{pij} \Delta y_{it-j} + \omega_{it} \tag{11}$$

The IPS test may be estimated in a single equation for each $i$ country. Then, the average of the calculated t-statistic from the individual equations of each $i$ country could be taken to validate the stationarity. After panel unit root testing, we apply the cointegration tests to validate the cointegration in the GCC panel. Kao [52] recommended the residual-based cointegration and suggested to test the unit root in the error-term generated from the fixed-effect model in the following way:

$$y_{it} = \alpha_t + \beta x_{it} + \psi_{it} \tag{12}$$

$$\Delta \widehat{\psi}_{it} = \rho \widehat{\psi}_{it-1} + v_{it} \tag{13}$$

If the residual series generated from Equation (12) is stationary using Equation (13), then we can conclude a cointegration in the model. Maddala and Wu [53] suggested another cointegration test based on the trace and maximum eigenvalue of Johansen [54]:

$$\lambda_{trace}(r) = -t \sum\nolimits_{i=r+1}^{n} \ln(1 - \widehat{\lambda}_i) \tag{14}$$

$$\lambda_{\text{maximum}}(r, r+1) = -t \ln(1 - \widehat{\lambda}_{r+1}) \tag{15}$$

$$-2 \sum\nolimits_{i=1}^{N} \log_{\psi}(\pi_i) \tag{16}$$

Equations (14) and (15) may be applied to each country, and the probability ($\pi_i$) value can be estimated. Then, the probability values would be accumulated to test the cointegration for a whole panel. Besides, we also applied the Pedroni [55] panel cointegration test based on 7 test statistics:

$$t_2 n \sqrt{n} Z_{\widehat{v}N,T} = t_2 n \sqrt{n} \Big| \sum_{i=1}^{n} \sum_{t=1}^{T} \hat{\psi}_{it-1}^2 / \widehat{L}_{11i}^2 \tag{17}$$

$$t_2 \sqrt{n} Z_{\widehat{\rho}N,T-1} = t_2 \sqrt{n} \Big[ \sum_{i=1}^{n} \sum_{t=1}^{T} 1/\widehat{L}_{11i}^2 (\hat{\psi}_{it-1}\Delta\hat{\psi}_{it} - \widehat{\lambda}_i] \Big| \sum_{i=1}^{n} \sum_{t=1}^{T} \hat{\psi}_{it-1}^2 / \widehat{L}_{11i}^2 \tag{18}$$

$$Z_{tN,T} = (\sum_{i=1}^{n} \sum_{t=1}^{t} 1/\widehat{L}_{11i}^2 (\hat{\psi}_{it-1}\Delta\hat{\psi}_{it} - \widehat{\lambda}_i) \Big| \sqrt{\widehat{\sigma}_{N,T}^2 \sum_{i=1}^{n} \sum_{t=1}^{t} \hat{\psi}_{it-1}^2 / \widehat{L}_{11i}^2} \tag{19}$$

$$Z_{tn,T^{-1}}^* = \sum_{i=1}^{n} \sum_{t=1}^{t} 1/\widehat{L}_{11i}^2 \hat{\psi}_{it-1}^* \Delta\hat{\psi}_{it}^* \Big| \sqrt{\widehat{S}_{n,t}^2 \sum_{i=1}^{n} \sum_{t=1}^{t} \hat{\psi}_{it-1}^2 / \widehat{L}_{11i}^2} \tag{20}$$

$$t\widetilde{Z}_{\widehat{\rho}n,t-1} / \sqrt{n} = t \sum_{i=1}^{n} (\hat{\psi}_{it-1}\Delta\hat{\psi}_{it} - \widehat{\lambda}_i) \Big| \sqrt{n} \sum_{i=1}^{n} \sum_{t=1}^{t} \hat{\psi}_{it-1}^2 \tag{21}$$

$$\widetilde{Z}_{tn,t-1} / \sqrt{n} = \sum_{i=1}^{n} (\hat{\psi}_{it-1}\Delta\hat{\psi}_{it} - \widehat{\lambda}_i) \Big| \sqrt{n}\widehat{\sigma}_i^2 \sum_{i=1}^{n} \sum_{t=1}^{t} \hat{\psi}_{it-1}^2 \tag{22}$$

$$\widetilde{Z}_{tn,t-1}^* / \sqrt{n} = \sum_{i=1}^{n} \hat{\psi}_{it-1}\Delta\hat{\psi}_{it} \Big| \sqrt{N}\widehat{S}_i^{*2} \sum_{i=1}^{n} \sum_{t=1}^{t} \hat{\psi}_{it-1}^{*2} \tag{23}$$

After establishing the cointegration, we utilize the Pooled Mean Group (PMG) of Pesaran et al. [56] for the long-run analyses:

$$\Delta BOT_{it} = \alpha_0 + \alpha_1 BOT_{it-1} + \alpha_2 POP_{it-1} + \alpha_3 NOP_{it-1} + \alpha_4 PRER_{it-1} + \alpha_5 NRER_{it-1}$$
$$+\sum_{j=1}^{n1-1} \delta_{1j}\Delta BOT_{it-j} + \sum_{j=0}^{n2-1} \delta_{2j}\Delta POP_{it-j} + \sum_{j=0}^{n3-1} \delta_{3j}\Delta NOP_{it-j}^2 + \sum_{j=0}^{n4-1} \delta_{4j}PRER_{it-j} \tag{24}$$
$$+\sum_{j=0}^{n5-1} \delta_{5j}\Delta NRER_{it-j} + \xi_{it}$$

Equation (24) can be estimated after the selection of optimum lag lengths chosen by the SIC. Then, Error Correction Term ($ECT_{it-1}$) can be replaced with the lagged-leveled variables to confirm the cointegration in the model. $\alpha_k$ may be normalized with the help of $\alpha_1$ to calculate the long-run coefficients. The estimates from PMG can also be verified using Pedroni's [57] methodology:

$$y_{it} = \alpha_i + \beta x_{it} + \psi_{it} \tag{25}$$

$$\hat{\beta}_{FMOLS} = \sum_{i=1}^{N} \Big[ \sum_{t=1}^{T} (x_{it} - \overline{x}_t)\hat{y}_{it}^+ + T\hat{\Delta}_{e\mu}^+ \Big] \Big| \sum_{i=1}^{N} \sum_{t=1}^{T} (x_{it} - \overline{x}_t)' \tag{26}$$

Equation (25) may be estimated after introducing the heterogeneity in the intercept. Then, the long-run effects through estimated $\hat{\beta}$ can be modified using Equation (26). Pedroni [57] solved the problem of endogeneity in Equation (26). Moreover, we may verify the results from PMG and fully modified Ordinary Least Square (OLS) by utilizing the Dynamic OLS (DOLS) of Kao and Chiang [58]. Dynamic OLS may be expressed in the following way:

$$y_{it} = \alpha_i + \beta x_{it} + \sum_{j=-1}^{j=+1} \Delta x_{i,t+j} + v_{it} \tag{27}$$

$$\hat{\beta}_{DOLS} = \sum_{i=t}^{N} (\sum_{t=1}^{T} z_{it}z'_{it})^{-1'} (\sum_{i=1}^{T} z_{it}\hat{z}_{it}^+) \tag{28}$$

Here, $z_{it} = [x_{it}, -\overline{x}_i, \Delta x_{i,t+q}, \ldots\ldots, x_{it}]$. Equation (27) would be regressed with lead and lag differenced independent variables and $\hat{\beta}$ could be normalized using Equation (28) to remove the endogeneity.

## 4. Results

At first, we test the possibility of dependency in the time series. We apply the BDS test for each series of each country with $H_0$: a time series is independently distributed. Saudi Arabia, Kuwait, and Oman share the same data on oil prices because of the homogenous period. Therefore, the oil prices' series is tested only once for Kuwait, and the testing is not repeated for KSA and Oman. The oil prices' series is tested for the rest of the countries separately. Table 2 shows that $H_0$ is rejected for all variables. Therefore, the investigated series are showing a nonlinear dependence and the nonlinear relationships between variables are expected.

**Table 2.** Broock–Dechert–Scheinkman (BDS) test.

| Country | Dimension Variable | 2 | 3 | 4 | 5 |
|---|---|---|---|---|---|
| Bahrain | $BOT_t$ | 0.0778 (0.0000) | 0.1228 (0.0000) | 0.1373 (0.0000) | 0.1358 (0.0000) |
| | $RER_t$ | 0.1558 (0.0000) | 0.2558 (0.0000) | 0.3280 (0.0000) | 0.3710 (0.0000) |
| | $OP_t$ | 0.1306 (0.0000) | 0.2229 (0.0000) | 0.3066 (0.0000) | 0.3509 (0.0000) |
| Kuwait | $BOT_t$ | 0.0797 (0.0000) | 0.1195 (0.0000) | 0.1247 (0.0000) | 0.1206 (0.0000) |
| | $RER_t$ | 0.1799 (0.0000) | 0.3005 (0.0000) | 0.3841 (0.0000) | 0.4365 (0.0000) |
| | $OP_t$ | 0.1600 (0.0000) | 0.2787 (0.0000) | 0.3639 (0.0000) | 0.4122 (0.0000) |
| Oman | $BOT_t$ | 0.0492 (0.0001) | 0.0683 (0.0007) | 0.0557 (0.0222) | 0.0423 (0.0913) |
| | $RER_t$ | 0.1678 (0.0000) | 0.2819 (0.0000) | 0.3594 (0.0000) | 0.4074 (0.0000) |
| Qatar | $BOT_t$ | 0.0904 (0.0000) | 0.1506 (0.0000) | 0.1659 (0.0000) | 0.2129 (0.0000) |
| | $RER_t$ | 0.2131 (0.0000) | 0.3641 (0.0000) | 0.4683 (0.0000) | 0.5349 (0.0000) |
| | $OP_t$ | 0.1523 (0.0000) | 0.2732 (0.0000) | 0.3580 (0.0000) | 0.4029 (0.0000) |
| Saudi Arabia | $BOT_t$ | 0.1191 (0.0000) | 0.1945 (0.0000) | 0.2386 (0.0000) | 0.2599 (0.0000) |
| | $RER_t$ | 0.1765 (0.0000) | 0.3024 (0.0000) | 0.3902 (0.0000) | 0.4492 (0.0000) |
| UAE | $BOT_t$ | 0.0590 (0.0000) | 0.1230 (0.0000) | 0.1389 (0.0000) | 0.1062 (0.0000) |
| | $RER_t$ | 0.1914 (0.0000) | 0.3309 (0.0000) | 0.4215 (0.0000) | 0.4788 (0.0000) |
| | $OP_t$ | 0.1647 (0.0000) | 0.2647 (0.0000) | 0.3088 (0.0000) | 0.3322 (0.0000) |

After testing the BDS, we corroborate the nonlinearity in the series. So, we split the $OP_t$ into $POP_t$ and $NOP_t$, and $RER_t$ into $PRER_t$ and $NRER_t$ using Shin et al. [47] methodology. After that, we apply the panel unit root tests. In Table 3, the results show that all variables are non-stationary at their levels and are stationary after first differencing. So, we may move towards panel cointegration testing.

**Table 3.** Unit-root analyses.

| Test | Series | Level-Variables | | Differenced-Variables | |
| --- | --- | --- | --- | --- | --- |
| | | C | C&T | C | C&T |
| LLC | $BOT_{it}$ | −1.3748 | 0.1086 | −4.2998 *** | −2.6179 *** |
| | $POP_{it}$ | −1.0957 | −0.8888 | −10.7994 *** | −9.7220 *** |
| | $NOP_{it}$ | 2.5129 | −1.1236 | −13.3411 *** | −12.0371 *** |
| | $PRER_{it}$ | 2.2623 | −1.4989 | −14.0542 *** | −12.1978 *** |
| | $NRER_{it}$ | −1.0902 | −0.9194 | −12.6662 *** | −10.9018 *** |
| IPS | $BOT_{it}$ | −0.9548 | −0.1214 | −11.7930 *** | −10.6834 *** |
| | $POP_{it}$ | 1.9755 | −0.3818 | −9.2387 *** | −7.9786 *** |
| | $NOP_{it}$ | 5.0301 | −0.4016 | −11.3402 *** | −10.1354 *** |
| | $PRER_{it}$ | 4.7012 | −0.1213 | −11.5483 *** | −10.7280 *** |
| | $NRER_{it}$ | −0.1329 | −0.7590 | −6.0569 *** | −5.3464 *** |

Note: *** shows stationarity at 1% level of significance. C is intercept and T is time trend.

In Table 4, the results of Pedroni's [55] panel cointegration show that five out of seven test statistics are showing the evidence of cointegration in the hypothesized panel model. Kao's [52] test also validates the cointegration in the model. Moreover, the results of the Maddala and Wu's [53] test corroborate four cointegration vectors in the Trace test and three cointegration vectors in the Max-Eigen test. Hence, this test also validates the conclusion of cointegration. The cointegration results corroborate that the oil prices, the real exchange rates, and trade balances are moving on the long-run path in a panel of the GCC countries. Here, it is interesting to compare this relationship among the GCC countries. So, country-specific cointegration is also tested.

**Table 4.** Panel cointegration results.

| **Pedroni's Test** | | | | |
| --- | --- | --- | --- | --- |
| **Test** | **Statistic** | ***p*-Value** | **Statistic** | ***p*-Value** |
| v-stat. | 1.6774 | 0.0467 | 1.3590 | 0.0871 |
| rho-stat. | −0.6750 | 0.2498 | −0.6135 | 0.2698 |
| PP-statistic | −2.0455 | 0.0204 | −1.8205 | 0.0343 |
| ADF-stat. | −2.3520 | 0.0093 | −2.0566 | 0.0199 |
| Group rho-stat istic | 0.5538 | 0.7101 | | |
| Group PP-stat istic | −2.7058 | 0.0034 | | |
| Group ADF-stat istic | −2.7118 | 0.0033 | | |
| **Kao's Test** | | | | |
| ADF-stat istic | −4.13449 | 0.0000 | | |
| **Johansen-Fisher's Test** | | | | |
| **Co-Integrated Vectors** | **Trace Statistic** | ***p*-Value** | **Max-Eigen Statistic** | ***p*-Value** |
| None | 110.60 | 0.0000 | 81.40 | 0.0000 |
| At-most 1 | 72.43 | 0.0000 | 53.51 | 0.0000 |
| At-most 2 | 32.60 | 0.0011 | 22.70 | 0.0304 |
| At-most 3 | 18.77 | 0.0943 | 18.40 | 0.1040 |
| At-most 4 | 13.11 | 0.3611 | 13.11 | 0.3611 |

Table 5 shows the long-run coefficients from PMG, Fully Modified OLS (FMOLS), and DOLS. $POP_{it}$ and $NOP_{it}$ have positive impacts on the $BOT_{it}$ in the GCC panel. It means that increasing oil prices are found helpful in improving the $BOT_{it}$ and decreasing oil prices could deteriorate the $BOT_{it}$ in the GCC countries. In asymmetry analyses, we find that the chi-square values (*p*-values) are 0.525 (0.4687), 1.4846 (0.2244) and 1.0138 (0.314). Hence, the $H_0$ of symmetry is not rejected and the positive symmetrical effects of $POP_{it}$ and $NOP_{it}$ are found on $BOT_{it}$. The estimated symmetrical effects of oil prices on the trade balances in the GCC countries' panel are in contrast to the findings

of Rafiq et al. [7], Baek et al. [13], and Baek and Kwon [12]. These studies found the asymmetrical effects of oil prices on the trade of oil-exporting countries. Our estimated symmetrical effect of oil prices on the trade balances may be claimed due to aggregation biasness in a whole panel. Hence, country-specific analyses may provide more insights into the relationship between oil prices and trade. Therefore, we conducted the time-series analyses and presented in Tables 6–8.

**Table 5.** Long-run effects in a panel of Gulf Co-operation Council (GCC) countries.

| Variable | PMG | FMOLS | DOLS |
|---|---|---|---|
| $POP_{it}$ | 0.4302 (0.0019) | 0.2708 (0.0268) | 0.4910 (0.0003) |
| $NOP_{it}$ | 0.5213 (0.0008) | 0.4064 (0.0050) | 0.5862 (0.0000) |
| $PRER_{it}$ | 0.6474 (0.0432) | 0.9672 (0.0039) | 1.1531 (0.0001) |
| $NRER_{it}$ | 0.5242 (0.0113) | 0.4470 (0.0138) | 0.7228 (0.0009) |

**Table 6.** Ng-Perron test.

| Country | Variable | Level | | | | First Difference | | | |
|---|---|---|---|---|---|---|---|---|---|
| | | MZa | MZt | MSB | MPT | MZa | MZt | MSB | MPT |
| Bahrain | $BOT_t$ | −11.011 | −2.277 | 0.207 | 8.618 | −18.337 ** | −3.028 ** | 0.165 ** | 4.971 ** |
| | $POP_t$ | −3.255 | −1.232 | 0.378 | 27.052 | −17.799 ** | −2.957 ** | 0.166 ** | 5.276 ** |
| | $NOP_t$ | −9.679 | −2.144 | 0.229 | 9.808 | −18.386 ** | −3.020 ** | 0.164 ** | 5.031 ** |
| | $PRER_t$ | −10.460 | −2.282 | 0.218 | 8.734 | −18.282 ** | −3.018 ** | 0.165 ** | 5.017 ** |
| | $NRER_t$ | −3.109 | −1.247 | 0.401 | 29.312 | −15.337 * | −2.766 * | 0.180 * | 5.962 * |
| Kuwait | $BOT_t$ | −12.634 | −2.513 | 0.199 | 7.213 | −22.915 ** | −3.378 ** | 0.147 ** | 4.021 ** |
| | $POP_t$ | −9.096 | −2.128 | 0.234 | 10.035 | −23.289 ** | −3.395 ** | 0.146 ** | 4.020 ** |
| | $NOP_t$ | −7.067 | −1.782 | 0.252 | 13.027 | −23.474 ** | −3.409 ** | 0.145 ** | 3.981 ** |
| | $PRER_t$ | −1.863 | −0.680 | 0.365 | 31.445 | −23.392 ** | −3.397 ** | 0.145 ** | 4.035 ** |
| | $NRER_t$ | −2.793 | −1.172 | 0.419 | 32.305 | −20.083 ** | −3.129 ** | 0.156 ** | 4.782 ** |
| Oman | $BOT_t$ | −11.241 | −2.358 | 0.210 | 8.171 | −22.323 ** | −3.339 ** | 0.150 ** | 4.093 ** |
| | $PRER_t$ | −9.200 | −2.105 | 0.229 | 10.062 | −23.397 ** | −3.408 ** | 0.146 ** | 3.968 ** |
| | $NRER_t$ | −3.361 | −1.247 | 0.371 | 26.163 | −22.591 ** | −3.357 ** | 0.149 ** | 4.059 ** |
| Qatar | $BOT_t$ | −5.482 | −1.603 | 0.292 | 16.455 | −22.587 ** | −3.485 ** | 0.142 ** | 4.125 ** |
| | $POP_t$ | −7.382 | −1.891 | 0.256 | 12.393 | −24.643 *** | −3.483 *** | 0.141 *** | 3.857 ** |
| | $NOP_t$ | −6.006 | −1.659 | 0.276 | 15.068 | −20.589 ** | −3.190 ** | 0.155 ** | 4.534 ** |
| | $PRER_t$ | −4.427 | −1.381 | 0.312 | 19.664 | −21.457 ** | −3.254 ** | 0.152 ** | 4.374 ** |
| | $NRER_t$ | −0.222 | −0.164 | 0.738 | 109.74 | −20.987 ** | −3.214 ** | 0.153 ** | 4.554 ** |
| Saudi Arabia | $BOT_t$ | −6.299 | −1.761 | 0.280 | 14.461 | −21.386 ** | −3.235 ** | 0.151 ** | 4.474 ** |
| | $PRER_t$ | −8.053 | −1.990 | 0.247 | 11.364 | −23.477 ** | −3.418 ** | 0.146 ** | 3.933 ** |
| | $NRER_t$ | −4.573 | −1.466 | 0.321 | 19.584 | −22.350 ** | −3.350 ** | 0.148 ** | 4.088 ** |
| UAE | $BOT_t$ | −7.662 | −1.948 | 0.254 | 11.910 | −20.462 ** | −3.191 ** | 0.156 ** | 4.499 ** |
| | $POP_t$ | −3.003 | −1.211 | 0.403 | 29.941 | −18.434 ** | −2.985 ** | 0.162 ** | 5.502 ** |
| | $NOP_t$ | −3.905 | −1.347 | 0.345 | 22.469 | −17.984 ** | −2.963 ** | 0.165 ** | 5.270 ** |
| | $PRER_t$ | −5.408 | −1.633 | 0.302 | 16.797 | −14.829 * | −2.698 * | 0.182 * | 6.280 * |
| | $NRER_t$ | −7.001 | −1.831 | 0.262 | 13.042 | −17.033 * | −2.860 * | 0.168 * | 5.683 * |

Note: *, ** and *** corroborate stationarity at 10%, 5% and 1%, respectively.

**Table 7.** Bound and diagnostic tests.

| Country | F-Statistics | Heteroscedasticity | Serial Correlation | Normality | Functional Form |
|---|---|---|---|---|---|
| Bahrain | 2.4736 | 0.9004 (0.5071) | 0.5493 (0.5833) | 3.0024 (0.2245) | 0.0187 (0.8920) |
| Kuwait | 3.9544 | 1.1080 (0.3744) | 0.3510 (0.7062) | 4.1245 (0.1247) | 2.0197 (0.1630) |
| Oman | 7.9131 | 0.5728 (0.8375) | 0.5220 (0.5981) | 0.7695 (0.6806) | 1.1303 (0.2952) |
| Qatar | 6.1550 | 0.9787 (0.5064) | 1.0986 (0.3704) | 0.2308 (0.8910) | 3.1689 (0.1027) |
| Saudi Arabia | 9.6230 | 1.3232 (0.2619) | 0.5704 (0.5703) | 0.5632 (0.7546) | 0.9348 (0.3399) |
| UAE | 8.1214 | 1.2704 (0.3606) | 2.8722 (0.1286) | 0.2739 (0.8720) | 0.1786 (0.6837) |

Critical Bound F-values, At 1% 3.0379–4.1121, At 5% 2.3851–3.3551, At 10% 2.0766–2.9892.

**Table 8.** Autoregressive Distributive Lag (ARDL) results.

| Country | Bahrain | Kuwait | Oman | Qatar | Saudi Arabia | UAE |
|---|---|---|---|---|---|---|
| Long-run | | | | | | |
| $POP_t$ | −0.0624 (0.8549) | −0.9230 (0.0074) | 0.3434 (0.0051) | −1.3938 (0.1289) | 0.5747 (0.0395) | 0.7912 (0.0100) |
| $NOP_t$ | 0.3664 (0.0086) | 0.4702 (0.2321) | 0.5773 (0.0087) | 3.7977 (0.0284) | −0.1496 (0.7527) | 1.5126 (0.0020) |
| $PRER_t$ | 2.1413 (0.0075) | 0.4743 (0.6909) | 0.6655 (0.0599) | 8.3092 (0.0502) | −0.1829 (0.8843) | 5.6208 (0.0028) |
| $NRER_t$ | −0.1304 (0.8754) | −2.1083 (0.0039) | 0.2901 (0.0568) | −1.9911 (0.1193) | 1.2179 (0.0013) | 1.6121 (0.0112) |
| Intercept | 0.1071 (0.8859) | 0.1934 (0.5697) | 0.0473 (0.5933) | 9.2719 (0.0123) | 1.4760 (0.0009) | −0.3957 (0.7035) |
| Short-run | | | | | | |
| $\Delta BOT_{t-1}$ | | | | | 0.2031 (0.0604) | |
| $\Delta POP_t$ | −0.0289 (0.8552) | 0.5654 (0.0044) | 0.8298 (0.0001) | 0.9698 (0.0050) | 0.6236 (0.0004) | 0.6035 (0.0095) |
| $\Delta POP_{t-1}$ | | | 0.7942 (0.0007) | −0.4241 (0.0402) | | |
| $\Delta NOP_t$ | 0.1696 (0.0579) | 0.2396 (0.2479) | 0.6419 (0.0000) | 1.3833 (0.0004) | 0.5098 (0.0428) | −0.0162 (0.9611) |
| $\Delta NOP_{t-1}$ | | | −0.2235 (0.0204) | | | |
| $\Delta PRER_t$ | 0.9912 (0.0395) | 0.2417 (0.6911) | 0.4166 (0.0218) | 2.1082 (0.0100) | −0.0816 (0.8852) | −0.2216 (0.8478) |
| $\Delta PRER_{t-1}$ | | | | −0.5441 (0.1416) | | |
| $\Delta NRER_t$ | −0.7438 (0.2263) | −1.0744 (0.0043) | 0.4848 (0.0546) | 0.8446 (0.1803) | 0.5435 (0.0122) | 1.2296 (0.0088) |
| $ECT_{t-1}$ | −0.4629 (0.0188) | −0.5096 (0.0048) | −0.6260 (0.0000) | −0.3642 (0.0123) | −0.4463 (0.0000) | −0.7628 (0.0005) |

The positive effects of $PRER_{it}$ and $NRER_{it}$ are found on the $BOT_{it}$ in the GCC panel. It means that the depreciation of the local currency helps to improve the $BOT_{it}$ in GCC countries and appreciation is found to be harmful for the $BOT_{it}$. Further, asymmetry is analyzed in the effects of $PRER_{it}$ and $NRER_{it}$. The symmetry is proved in PMG results with chi-square value = 0.1766 and *p*-value = 0.6749 in the Wald test. However, the asymmetrical effects of $PRER_{it}$ and $NRER_{it}$ are found in the FMOLS and DOLS with the estimated 3.0003 (0.0832) and 2.9683 (0.0849) of chi-square value (*p*-value), respectively.

Moreover, the effect of depreciation ($PRER_{it}$) is found to be larger than that of the effect of appreciation ($NRER_{it}$) in all estimates. Mahmood et al. [59] also reported the asymmetrical effects of the exchange rates on trade in Saudi Arabia. We find mixed evidence of symmetry and asymmetry in the relationship between real exchange rates and trade balances. Hence, we need to investigate this issue in more detail. Therefore, we conduct the time series analyses which are presented in Tables 7 and 8.

After the panel estimates, our objective is to test the effects of oil prices and real exchange rates on trade in each country to avoid the expected aggregation biasness in the panel's estimates. At first, we apply the Ng-Perron test, with both intercept (C) and time (T) specification, to verify the unit root problem in the individual series, i.e. $BOT_t$, $POP_t$, $NOP_t$, $PRER_t$, and $NRER_t$. The testing of $POP_t$ and $NOP_t$ is not repeated for Oman and KSA, which have been tested in the case of Kuwait. Table 6 shows that all tested series are non-stationary at their levels and stationary at their first differences. So, the order of integration is one and we may move to the cointegration analyses.

After the selection of optimum lag length, Table 7 shows the results of the bound test, which has been applied on the function $F(BOT_t/POP_t, NOP_t, PRER_t, NRER_t)$. We also apply the diagnostic tests on the estimated models. We find that the p-values are more than 0.10 in the case of all diagnostic tests. Hence, the estimated that models are statistically reliable and we may move to the bound testing procedures. We utilize the F-statistics from Kripfganz and Schneider [60] which are efficient in the case of a small sample. The results show that the estimated F-statistic from the model of Bahrain is very low and we could not validate a cointegration. However, the cointegration can alternatively be tested with the coefficient of $ECT_{t-1}$ [50]. The coefficient of $ECT_{t-1}$ is negative and statistically significant in the model of Bahrain at a 5% level of significance, presented at the bottom of Table 8. So, the cointegration is proved for Bahrain. The estimated F-statistic from the model of Kuwait is found to be greater than the upper bound statistic at a 5% level of significance. Hence, the model of Kuwait is integrated in the long run. The estimated F-values from the models of Oman, Qatar, Saudi Arabia, and the UAE are found to be larger than the upper bound value at 1%. Hence, the cointegration is proved in all countries' models. The results of cointegration corroborate that variables of the hypothesized model are moving on a long-run path. Moreover, the trade balances are sufficiently explained by the oil prices and real exchange rates in all countries' models.

In Table 8, $POP_t$ has a positive impact on $BOT_t$ in Oman, Saudi Arabia, and the UAE in the long run. Hence, increasing oil prices improved the trade balances in these countries. Rafiq et al. [7] also reported a positive effect of increasing oil prices on the oil–trade balances in oil exporting countries. Moreover, Timilsina [3] argued that increasing oil prices would increase the total exports of oil-intensive nations. It shows that the percentage decline in oil demand is lesser than the percentage increase in the oil prices, and revenues from oil exports are increasing in Oman, Saudi Arabia, and the UAE. Further, increasing oil prices may also increase income and imports in oil-exporting countries. Hence, the results corroborate that increasing oil exports are larger than the increasing imports. The effect of $POP_t$ is found to be negative in Kuwait and insignificant in Bahrain and Qatar. The economy of Kuwait is heavily dependent on the oil sector in terms of income and exports. Oil exports are expected to increase with a rise in oil prices due to a price-inelastic demand. On the other hand, imports are expected to increase due to increasing income because of increasing oil prices. Korhonen and Ledyaeva [8] corroborated that increasing oil prices were found to be responsible for increasing imports in oil-producing countries. The negative effect of increasing oil prices on the trade balances corroborates that increasing imports' payments are more than increasing exports' revenues in Kuwait. This result is in the line with the theoretical predictions of Le and Chang [6]. Le and Chang [6] argued that increasing oil prices might reduce the income and purchasing power of oil-importing countries and could reduce the demand for products from oil-exporting countries. In the short run, $POP_t$ has a positive impact on the $BOT_t$ of all GCC countries except Bahrain. So, increasing oil prices helped to improve the trade balances in most GCC countries.

In the long run, $NOP_t$ has a positive effect on the $BOT_t$ in Bahrain, Oman, Qatar, and the UAE. Hence, decreasing oil prices worsened trade balances. The argument of low price-elasticity of oil

demand is found to be valid in the case of decreasing oil prices and oil export revenues fall with the fall in oil prices. It means that a percentage increase in oil demand is lesser than a percentage decrease in oil prices. Hence, revenues from oil exports are decreasing in Bahrain, Oman, Qatar, and the UAE. This finding is contradicting the results of Rafiq et al. [7] who found that decreasing oil prices helped to improve the oil–trade balances in exporting countries. The effect of $NOP_t$ is found to be insignificant in Kuwait and Saudi Arabia. Moreover, we test the asymmetry in the long-run relationship between oil prices and the trade balance. The effects of $POP_t$ and $NOP_t$ are found to be asymmetrical in the case of Bahrain, Kuwait, Qatar, and Saudi Arabia due to mixed evidence of statistically significant and insignificant effects of $POP_t$ and $NOP_t$. On the other hand, the effects of $POP_t$ and $NOP_t$ on $BOT_t$ are positive in Oman and the UAE. The chi-square (p-value) of the Wald test is found to be 5.6086 (0.0235) and 6.9024 (0.0275) in Oman and the UAE models, respectively. Hence, the $H_0$ of symmetry is rejected and the magnitude of the effects of $POP_t$ and $NOP_t$ are found statistically different. The long-run asymmetry in the relationship of oil prices and trade is also corroborated in the literature [7,12,13]. In the short run, the effect of $NOP_t$ is found to be positive in all GCC countries except Kuwait and UAE. The short-run effects of $POP_t$ and $NOP_t$ are found asymmetrical in the case of Bahrain, Kuwait, and the UAE due to mixed evidence of statistically significant and insignificant effects of $POP_t$ and $NOP_t$. In the rest of the cases, symmetry is corroborated with 0.8789 (0.3544) in Saudi Arabia, and asymmetry is corroborated with 11.6003 (0.0007) and 17.8725 (0.0001) in Oman and Qatar, respectively.

In the long run, real depreciation of the local currency ($PRER_t$) has a positive effect in Bahrain, Oman, Qatar, and the UAE. It means that depreciation helped to improve trade balances because of increasing exports and/or decreasing imports. Hence, depreciation is found to be helpful in improving the trade balances in Bahrain, Oman, Qatar, and the UAE. The effect of the real appreciation of the local currency ($NRER_t$) is found to be positive in Oman, the UAE, and Saudi Arabia. So, appreciation deteriorated the trade balances because of decreasing exports and/or increasing imports. Hence, appreciation worsened the trade balances in Oman, the UAE, and Saudi Arabia. The effect of $NRER_t$ is found to be negative on $BOT_t$ in Kuwait. Hence, appreciation helped Kuwait to improve its trade balance. The oil sector contributes a major proportion to the total exports of Kuwait, and appreciation policy is expected to improve oil exports' revenues because of the price-inelastic demand for oil in the world market. On the other hand, appreciation is supposed to depress the imports' payments. Hence, both increasing exports and decreasing imports are helping the Kuwaiti economy to improve the trade balance in the times of the appreciation policy. The asymmetry in the relationship of the exchange rates and the trade balances is corroborated because of mixed evidence of statistically significant and insignificant effects of $PRER_t$ and $NRER_t$ in Bahrain, Kuwait, Qatar, and Saudi Arabia. $PRER_t$ and $NRER_t$ have positive effects on $BOT_t$ in Oman and the UAE. The chi-square values (*p*-values) from Oman and UAE models are found as 2.9132 (0.0960) and 6.9357 (0.0084), respectively. Hence, the $H_0$ of symmetry is rejected in the case of both countries. Asymmetry is corroborated in the relationship of the real exchange rates and trade balances in all GCC countries. Mahmood et al. [59] also reported the asymmetrical effect of the exchange rates on the service sector's exports in Saudi Arabia.

In the short run, $PRER_t$ has a positive impact on the trade balances in Bahrain, Oman, and Qatar. It means that depreciation helped to improve the trade balances of these countries in the short run. The effect of appreciation ($NRER_t$) is found to be positive in Oman, Saudi Arabia, and the UAE. It means that appreciation worsened the trade balances in these countries. The effect of $NRER_t$ is found to be negative in Kuwait. Hence, appreciation improved the trade balance in Kuwait. The asymmetrical effects of $PRER_t$ and $NRER_t$ are validated in all the countries, except Oman. In Oman, the chi-square value (*p*-value) from the Wald test is found as 0.1657 (0.6840) and we accept the $H_0$ of symmetry. So, the effects of both appreciation and depreciation on the $BOT_t$ are found positive and symmetrical in Oman.

## 5. Conclusions

Literature has investigated the effect of oil prices on the stock markets and other macroeconomic performances of the GCC countries. The recent literature is silent towards the effect of oil prices on the trade balances in the GCC countries. This study has investigated this issue and also tested the presence of nonlinearity. The BDS test shows that hypothesized variables are not independently distributed. So, we apply the nonlinear effects of the oil prices and real exchange rates on the trade balances in the GCC countries. We divide the oil prices and real exchange rate series into increasing and decreasing series to conduct the asymmetry analyses. The panel unit root tests corroborate that all variables are the first-differenced stationary, and panel cointegration tests have validated the cointegration in the hypothesized panel model. Then, we apply the PMG, FMOLS, and DOLS on the GCC panel. The results suggest that both rising and falling oil prices have positive and symmetrical impacts on the trade balances in the GCC countries. The effects of rising and falling real exchange rates are positive and asymmetrical on the trade balances, and the effect of depreciation is found to be larger than that of the effect of appreciation.

After the panel estimations, we perform the time series' analyses on each country's model to avoid the possible panel aggregation biasness. We apply the unit root test on time series variables and find that all variables are the first-differenced stationary. Moreover, cointegration is also established in all countries' models. We corroborate that rising oil prices have a positive effect on the trade balances in Oman, Saudi Arabia, and the UAE, and a negative effect in Kuwait. Hence, the price-inelastic oil demand is corroborated in Oman, Saudi Arabia, and the UAE. It means that increasing oil prices helped to improve the trade balances because of increasing exports' revenue and/or decreasing imports' payment. On the other hand, increasing oil prices depressed the trade balance in Kuwait. The income of Kuwait is highly dependent on the oil sector and increasing oil prices may raise the national income. The increasing income would accelerate the imports which may deteriorate the trade balance. On the other hand, the oil sector also carries a significant proportion of exports. Then, oil exports' revenue may increase with rising oil prices due to inelastic demand. Our results corroborate that increasing imports' payments are more than increasing exports' revenues in times of increasing oil prices in Kuwait. The decreasing oil prices have a positive effect on the trade balances in Bahrain, Oman, Qatar, and the UAE and have an insignificant effect in Kuwait and Saudi Arabia. In the short run, rising oil prices have a positive effect on the trade balances in all countries except Bahrain. The effect of decreasing oil prices is found to be positive in all countries except Kuwait and the UAE. Moreover, asymmetry in the short- and long-run relationships of oil prices and trade balances is corroborated in all GCC countries except Saudi Arabia.

The depreciation of the local currency helped to improve the trade balances in Bahrain, Oman, Qatar, and the UAE in the long run. It means that depreciation assisted to increase exports and/or to decrease imports in these countries. The depreciation policy may decrease the price of products in the international market and demand for the product may rise. Further, increasing demand should be more than the decreasing prices to ensure a rise in exports' revenue. The fact that depreciation improved the trade balances in Bahrain, Oman, Qatar, and the UAE is corroborated. The depreciation may also result in local inflation in the country which may decrease the real value of money. Moreover, the depreciation may increase the price of imports which may decrease the demand for imports. Resultantly, imports' payment may be declined which may improve the trade balances resultantly. Therefore, we recommend the depreciation policy to Bahrain, Oman, Qatar, and UAE. The appreciation deteriorated the trade balances in Oman, the UAE, and Saudi Arabia. It means that the appreciation policy increased the price of exports and decreased the price of imports in the international market. Resultantly, the demand for exports declined and demand for imports raised. Hence, appreciation is found to be harmful for the trade balances and we recommend Oman, the UAE, and Saudi Arabia to avoid the appreciation policy. Appreciation improved the trade balance in Kuwait. It may be claimed due to the reason that the major exports of Kuwait are from the oil sector which is price inelastic. Hence, appreciation could not discourage a significant quantity of exports and improved the

exports' revenues resultantly. Therefore, the appreciation policy is recommended to Kuwait which has the potential to improve the trade balance. The asymmetrical long-run relationship between real exchange rates and the trade balances is observed in all investigated GCC countries. Hence, the GCC countries may utilize the estimated asymmetrical coefficients of depreciation and appreciation to predict the expected changes in the trade balances in case of any exchange rate policy. In the short run, depreciation helped to improve the trade balances in Bahrain, Oman, and Qatar. Therefore, a depreciation policy is suggested to these countries to improve their short-run trade balances. In the short run, the appreciation worsened the trade balances in Oman, Saudi Arabia, and the UAE. Hence, these countries should avoid the appreciation policy in the short run to avoid deterioration in the trade balances. The asymmetry in the short run relationship of the exchange rates and trade balances is validated in the GCC countries except for Oman. A summary of long-run results is reported in Figure 2 and a summary of country-specific policy implications is displayed in Table 9.

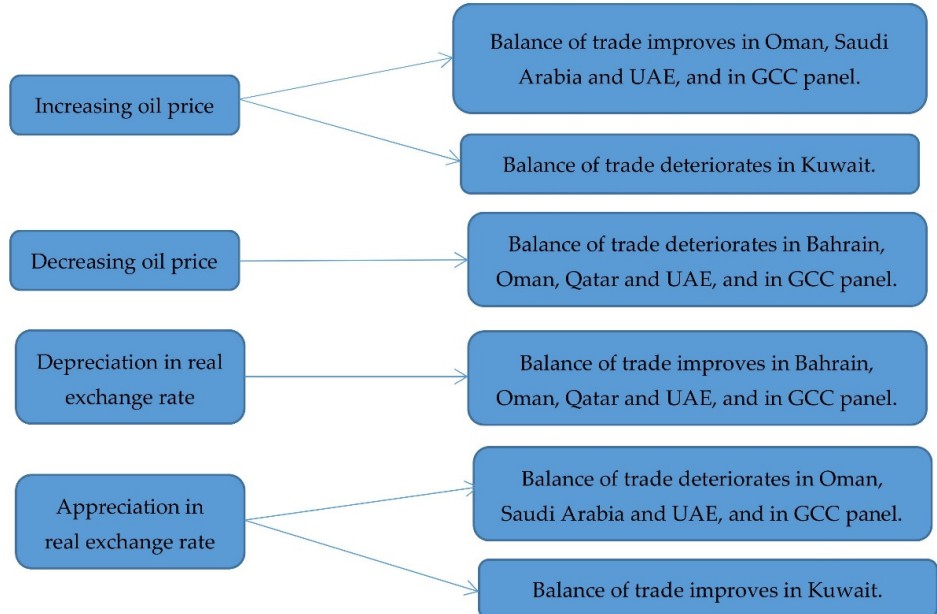

**Figure 2.** Long-run results.

**Table 9.** Summary of country-specific policy implications.

| Country | Major Finding | Suggested Policy | Expected Policy Outcome |
|---|---|---|---|
| Bahrain | Depreciation improves the trade balance. | To depreciate the local currency | Exports may increase and/or imports may decrease, and the trade balance may improve. |
| Kuwait | Appreciation improves the trade balance. | To appreciate the local currency | Exports may increase and/or imports may decrease, and the trade balance may improve. |
| Qatar | Depreciation improves the trade balance. | To depreciate the local currency | Exports may increase and/or imports may decrease, and the trade balance may improve. |
| Oman | Depreciation improves the trade balance. | To depreciate the local currency | Exports may increase and/or imports may decrease, and the trade balance may improve. |
| | Appreciation deteriorates the trade balance. | To avoid appreciation policy | The trade balance would be protected from deterioration. |
| Saudi Arabia | Appreciation deteriorates the trade balance. | To avoid appreciation policy | The trade balance would be protected from deterioration. |
| UAE | Depreciation improves the trade balance. | To depreciate the local currency | Exports may increase and/or imports may decrease, and the trade balances may improve. |
| | Appreciation deteriorates the trade balance. | To avoid appreciation policy | The trade balance would be protected from deterioration. |

　　　We face the constraints of limited data availability in terms of period and the choice of proxy of the exchange rates. Future research may extend the scope of the study by collecting a greater time frequency of data and by collecting the real effective exchange rates data requesting the central banks of GCC countries. Moreover, the analyses of sectoral trade data in the hypothesized model may provide more detailed insights into the sectoral trade balances due to any change in oil prices and exchange rates.

**Author Contributions:** Conceptualization, T.T.Y.A. and H.M.; Methodology, T.T.Y.A.; Software, H.M.; Validation, T.T.Y.A.; Formal Analysis, H.M.; Investigation, T.T.Y.A.; Resources, T.T.Y.A.; Data Curation, H.M.; Writing-Original Draft Preparation, T.T.Y.A.; Writing-Review & Editing, H.M.; Visualization, T.T.Y.A.; Supervision, T.T.Y.A.; Project Administration, T.T.Y.A.; Funding Acquisition, T.T.Y.A. All authors have read and agreed to the published version of the manuscript.

**Funding:** This research was funded by the Deanship of Scientific Research at Prince Sattam bin Abdulaziz University Alkharj, Saudi Arabia, grant number 2020/02/16618.

**Acknowledgments:** We acknowledge the anonymous referees for their useful comments.

**Conflicts of Interest:** The authors declare no conflict of interest.

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
