# Peer review of "The Oil Price and Trade Nexus in the Gulf Co-Operation Council Countries"

_resources, doi:10.3390/resources9120139_

Round 1
Reviewer 1 Report
An excellent piece of research with very interesting confirmatory results, however there are not enough economic comments and comparisons with previous studies, what significantly downgrade que quality of the investigation and applications in real markets. Specificaly two major questions remain unanswered in this sense:
- Line 292 onwards. The anlaysis of cointegration is very interesting but results seem to be vague and messy. It is important to be concise in the cointegration relationships, how many?, which are the countries behaving as one regarding long term dynamics?. Are there any economic reasoning supporting these considerations?. This will help to better understand implications of cointegration issues.
- Line 407. Why OP price increase has a negative impact in Kuwait?. This result is somehow contradictory and an explanation should be proposed. Similarly in line 418, depreciation of the local currency has benefits for the Balance of Trade in Kuwait. Please explain and support with relevant information.
Author Response
Reviewer 1
Comment:
An excellent piece of research with very interesting confirmatory results, however, there are not enough economic comments and comparisons with previous studies, what significantly downgrade the quality of the investigation and applications in real markets. Specifically, two major questions remain unanswered in this sense:
Response:
Thank you very much for your appreciation and all of your valuable comments which really help us to improve the quality of the paper.
Our results are now compared with the most important studies on the topic in lines 353-355, 366-369, 399-402, 406-415, 419-424, 430-431, and 454-456. Particularly, the economic arguments of literature are incorporated with the different results of Kuwait from others in lines 406-415 and 444-449.
Comment:
Line 292 onwards, the analysis of cointegration is very interesting but results seem to be vague and messy. It is important to be concise in the cointegration relationships, how many? Which are the countries behaving as one regarding long term dynamics? Is there any economic reasoning supporting these considerations? This will help to better understand the implications of cointegration issues.
Response:
The discussions on cointegration results are extended in lines 337-345 for panel cointegration results and in lines 386-396 for time series cointegration results.
Comment:
Line 407, why OP price increase has a negative impact in Kuwait? This result is somehow contradictory and an explanation should be proposed. Similarly, in line 418, the depreciation of the local currency has benefits for the Balance of Trade in Kuwait. Please explain and support with relevant information.
Response:
The justification for a negative relationship between OP and BOT of Kuwait is discussed in lines 406-415. The depreciation has statistically insignificant effects on the BOT in Kuwait. The mentioned unusual relationship in this comment is of appreciation and BOT. The justification for this relationship is discussed in lines 444-449.

Reviewer 2 Report
I found the topic interesting. However, there are three areas that need to be improved for this article to be publishable. They are listed as follows.
- The literature review is confusing and difficult to follow. It is basically a description the results obtained by related articles. What we really need is a more abstract debate that integrates relevant articles to generate a narrative that follows a logical sequence of ideas. That is, a literature review is not really a list of articles, it is a debate. It is a narrative that discusses different points of views, different angles of a problem, etc. For example, I can explain that a number of researchers have found a positive relationship between PO and BOT, and then I can suggest some possible explanations based on the findings of these researchers.
- The discussion is poor. The paper puts too much weight on results, but very little on the discussion part. For example, what are the implications of the findings? How are they related top previous work? How theoretical explanations may explain differences and similarities between countries?
- But the most problematic issue in this paper is the concept of real exchange rate used by the authors. They define real exchange rate as “the number of currency of i GCC country to buy one US dollar”. This is not correct. The definition is “the product of the nominal exchange rate and the ratio of prices between the two countries”. Actually, the definition given by the author corresponds to the “nominal exchange rate”. According to the economic theory, exports and imports respond to the real exchange rate, and not to the nominal exchange rate. Of course, the latter can influence the real exchange rate. But this is not necessarily the case because this also depends on differences in inflation rate between countries. This means that the results obtained from the empirical model are not robust and may be biased because they are based on a theoretical mistake.
Author Response
Reviewer 2
I found the topic interesting. However, there are three areas that need to be improved for this article to be publishable. They are listed as follows.
Comment:
The literature review is confusing and difficult to follow. It is basically a description of the results obtained by related articles. What we really need is a more abstract debate that integrates relevant articles to generate a narrative that follows a logical sequence of ideas. That is, a literature review is not really a list of articles, it is a debate. It is a narrative that discusses different points of views, different angles of a problem, etc. For example, I can explain that a number of researchers have found a positive relationship between PO and BOT, and then I can suggest some possible explanations based on the findings of these researchers.
Response:
Thank you very much for all of your appreciation and valuable comments which really help us to improve the quality of the paper.
Following this comment and comment No. 9 of Reviewer 3, we added a table in line 208 to classify the literature in (1) symmetrical and (2) asymmetrical relationship between OP and trade relationship and (3) GCC literature on OP and any macroeconomic performance.
The literature review has been improved and highlighted the nature of analysis in OP and trade relationships to discuss the major change in the OP-trade relationship chronologically. Then, the studies investigating the OP-any macroeconomic performances are discussed in GCC as the literature on a relationship between OP and trade relationship is missing in a particular case of GCC countries.
The logical sequence of literature is summarized in the last paragraph to trace the gap and contribution of the study.
Overall write-up of the literature review section is improved for a better readership.
Comment:
The discussion is poor. The paper puts too much weight on results, but very little on the discussion part. For example, what are the implications of the findings? How are they related to top previous work? How theoretical explanations may explain differences and similarities between countries?
Response:
Our results are now compared with the most important studies on the topic in lines 353-355, 366-369, 399-402, 406-415, 419-424, 430-431, and 454-456.
With help of economic arguments, the explanations of different directions of the effect of investigated variables are extended in line 406-415 and 444-449.
Implications are discussed in lines 497-514.
The overall discussions in this section are enhanced and improved.
Comment:
But the most problematic issue in this paper is the concept of the real exchange rate used by the authors. They define the real exchange rate as “the number of currency of i GCC country to buy one US dollar”. This is not correct. The definition is “the product of the nominal exchange rate and the ratio of prices between the two countries”. Actually, the definition given by the author corresponds to the “nominal exchange rate”. According to the economic theory, exports and imports respond to the real exchange rate, and not to the nominal exchange rate. Of course, the latter can influence the real exchange rate. But this is not necessarily the case because this also depends on differences in the inflation rate between countries. This means that the results obtained from the empirical model are not robust and may be biased because they are based on a theoretical mistake.
Response:
We utilized the real exchange rate in all estimations. But, in the previous version of this paper, we focused on the definition of the only nominal exchange rate by a mistake (to make clear that 1 US dollar = number of i GCC local currency and positive movement may show a depreciation and vice versa, the discussion was focus only to make clear the definition of PRER and NRER) and we forget to define the procedure to convert the nominal into the real exchange rate. Now, this issue is cleared in lines 229-233. Moreover, REER is ignored in the estimations due to the non-availability of data for all sample countries and periods.

Reviewer 3 Report
avoid too many acronyms at least in the abstract that is extremely difficult to follow.
Make reference to the Dutch disease (resource curse) See Collier 2007 The Bottom Billion ...
In the Introduction, a simple flow chart may be included, showing the impact of Oil Price on RER and BOT in the investigated countries. This chart should be linked to the research question.
The reasoning behind the paper is very intricated and uneasy to follow. Yoy should use consequential numbering to outline the different steps, t least in the introduction (lines 51 ...)
line 38 because of the reason that
lines 70-71 present present avoid repetition
While countries behave differently? Iran line 88 Saudi Arabia 89 ...
line 149 probed and found had
the literature presentation is very technical and uneasy to follow. Think about a synoptic table subdividing the literature according to the different streams / findings...
lines 178-180: describe more analytically the literature gaps
a real Discussion is missing
line 393 this present study
in the conclusion, authors should:
- introduce tips for further research
- indicate why their findings are useful and which are the practical implications. Why is the paper worth reading?
- the listing of the main findings is not sufficient
Explain why this study differs from
Oil Price and Energy Depletion Nexus in GCC Countries: Asymmetry Analyses TTY Alkhateeb, H Mahmood - Energies, 2020 - mdpi.com
quote further studies; e.g.
Oil price shocks and energy consumption in GCC countries: a system-GMM approach MI Haque - Environment, Development and Sustainability, 2020 - Springer
A multiple and partial wavelet analysis of the oil price, inflation, exchange rate, and economic growth nexus in Saudi Arabia
C Aloui, B Hkiri, S Hammoudeh… - … Finance and Trade, 2018 - Taylor & Francis
Oil price volatility, Islamic financial development and economic growth in Gulf Cooperation Council (GCC) countries
K Gazdar, MK Hassan, MF Safa, R Grassa - Borsa Istanbul Review, 2019 - Elsevier
Am, Muhammad Ashiq and Shanmugasundaram, G., Nexus between Crude Oil Price, Exchange Rate and Stock Market: Evidence from Oil Exporting and Importing Economies (January 2, 2017). International Journal of Humanities and Management Sciences, Volume 5, Issue 1 (2017)
Linkage between international trade and economic growth in GCC countries: Empirical evidence from PMG estimation approach
J Jouini - … journal of international trade & economic development, 2015 - Taylor & Francis
Modelling oil price-inflation nexus: The role of asymmetries AA Salisu, KO Isah, OJ Oyewole, LO Akanni - Energy, 2017 - Elsevier
The nexus of trade-weighted dollar rates and the oil prices: an asymmetric approach A Hatemi-J, Y El-Khatib - Journal of Economic Studies, 2020 - emerald.com
Improving the Forecast Accuracy of Oil-Stock Nexus in GCC Countries OI Nnachi - Theoretical Economics Letters, 2018 - scirp.org
check the English using, for instance, grammarly.com
Author Response
Reviewer 3
Comment:
Avoid too many acronyms at least in the abstract that is extremely difficult to follow.
Response:
Thank you very much for all of your valuable comments which really help us to improve the quality of the paper.
All acronyms are removed from the abstract and reduced in the other sections of this study.
Comment:
Make reference to the Dutch disease (resource curse) See Collier 2007 The Bottom Billion ...
Response:
The discussion on the resource curse is added using two references of The Dutch Disease (1977) and Collier (2007) in the introduction section in lines 46-49.
Comment:
In the Introduction, a simple flow chart may be included, showing the impact of Oil Price on RER and BOT in the investigated countries. This chart should be linked to the research question.
Response:
Figure 1 has been added in lines 75-92.
Comment:
The reasoning behind the paper is very intricate and uneasy to follow. You should use consequential numbering to outline the different steps, at least in the introduction (lines 51 ...)
Response:
The linking sentence from previous paragraphs “The above discussions argued that increasing OP might harm or support the BOT” has been changed in lines 54-55. Now, the arguments are discussed with the help of references to make it reader-friendly.
The discussions on the literature gap and contribution are extended in introduction 66-105 and literature review in line 209-218.
Comment:
Line 38 “because of the reason” that
Response:
Lines 38-39 are rephrased to avoid confusion and the statement “because of the reason that” is removed as per comment.
Comment:
Lines 70-71 present present avoid repetition
Response:
The duplication is removed by rephrasing in line 100.
Comment:
While countries behave differently? Iran line 88 Saudi Arabia 89 ...
Response:
These are results of Baek et al. (2019) which corroborated symmetry in Saudi Arabia and Asymmetry in Iran. This explanation is added in lines 121-122.
Comment:
Line 149 probed and found had
Response:
A typo mistake of “had” is removed in line 181.
Comment:
The literature presentation is very technical and uneasy to follow. Think about a synoptic table subdividing the literature according to the different streams/findings...
Response:
A table is added in line 208 which is summarized the main findings of the studies on the topic. We now discuss and classify to (1) symmetrical and (2) asymmetrical testing of OP-trade relationship and (3) the studies on testing OP-related studies in GCC countries.
Comment:
Lines 178-180: describe more analytically the literature gaps, a real Discussion is missing
Response:
An analytical discussion on the literature is summarized to trace the gap and contribution of the study in lines 209-218.
Comment:
Line 393 this “present” study
Response:
“present” is removed as per comment in line 470.
Comment:
In the conclusion, the authors should:
Introduce tips for further research
Indicate why their findings are useful and which the practical implications are. Why is the paper worth reading?
The listing of the main findings is not sufficient
Response:
The implications from the results of the study are extended in lines 497-514.
The possible future research is suggested in lines 515-520.
The discussion on the main findings is enhanced.
Comment:
Explain why this study differs from:
Oil Price and Energy Depletion Nexus in GCC Countries: Asymmetry Analyses TTY Alkhateeb, H Mahmood - Energies, 2020 - mdpi.com
quote further studies; e.g.
Oil price shocks and energy consumption in GCC countries: a system-GMM approach MI Haque - Environment, Development and Sustainability, 2020 - Springer
A multiple and partial wavelet analysis of the oil price, inflation, exchange rate, and economic growth nexus in Saudi Arabia
C Aloui, B Hkiri, S Hammoudeh… - … Finance and Trade, 2018 - Taylor & Francis
Oil price volatility, Islamic financial development and economic growth in Gulf Cooperation Council (GCC) countries
K Gazdar, MK Hassan, MF Safa, R Grassa - Borsa Istanbul Review, 2019 - Elsevier
Am, Muhammad Ashiq and Shanmugasundaram, G., Nexus between Crude Oil Price, Exchange Rate and Stock Market: Evidence from Oil Exporting and Importing Economies (January 2, 2017). International Journal of Humanities and Management Sciences, Volume 5, Issue 1 (2017)
Linkage between international trade and economic growth in GCC countries: Empirical evidence from PMG estimation approach
J Jouini - … journal of international trade & economic development, 2015 - Taylor & Francis
Modelling oil price-inflation nexus: The role of asymmetries AA Salisu, KO Isah, OJ Oyewole, LO Akanni - Energy, 2017 - Elsevier
The nexus of trade-weighted dollar rates and the oil prices: an asymmetric approach A Hatemi-J, Y El-Khatib - Journal of Economic Studies, 2020 - emerald.com
Improving the Forecast Accuracy of Oil-Stock Nexus in GCC Countries OI Nnachi - Theoretical Economics Letters, 2018 - scirp.org
Response:
All mentioned references are added in the last two paragraphs of the introduction section and the literature gap is discussed in lines 66-105.
Comment:
Check the English using, for instance, grammarly.com
Response:
The overall write-up is cared for and improved in all sections of the revised manuscript. Grammarly is also utilized to remove English mistakes.

Round 2
Reviewer 2 Report
I can see a significant improvement of the article. And I am please to see that the real exchange rate has been defined correctly. I think the article is ready for publication.
Author Response
Comments:
I can see a significant improvement in the article. And I am pleased to see that the real exchange rate has been defined correctly. I think the article is ready for publication.
Response:
Thank you very much for your recommendations.
Reviewer 3 Report
line 12: symmetric positive asymmetrical positive explain what it means ...
lines 46 - 48 are not clear: explained a phenomenon that .... rephrase the English.
ref. 9 should be eliminated; it is not a scientific source; Collier is enough
figure 1 - do not include acronyms (but them in brackets after the full wording
lines 95-97: where is the verb?
line 98 though
line 103 why there is a question mark ?
acronyms are still difficult to follow, even in the text; for instance, is it really necessary to abbreviate OP for oil price ? I would strongly suggest eliminating all acronyms; this operation just takes a few minutes.
You may also replace GCC in the title with "Gulf" then You explain in the introduction what GCC really stands for
line 209 - signified: rephrase
line 216 - we are highlighted motivated: rephrase this "excessive" sentence
229 - what is "number" of currency?
337 - Pedroni - quote
367 finding Mahmood : rephrase
400 [7] asymmetrical: sentence not grammatically correct
431 delete past
515-516: do not over-repeat the word "study"
a real discussion is still missing ...
You may include a further figure at the end, mirroring figure 1, and explaining the results
make a further effort to highlight the practical implications of your study
You may add at the end a table with a summary of the policy implications for each examined country
Author Response
Reviewer 3
Thank you very much for your comments which tremendously help us to improve the present version of article.
Comments:
Line 12: symmetric positive asymmetrical positive explain what it means...
Response:
Correction of asymmetric is done in line 13. The explanations about the meaning of asymmetric and symmetric are added in lines 14-16.
Comments:
Lines 46 - 48 are not clear: explained a phenomenon that .... rephrase the English.
Response:
The explanations are added in lines 48-50 for the linkages of appreciation and its effect on products’ price and demand.
Comments:
Ref. 9 should be eliminated; it is not a scientific source; Collier is enough
Response:
Reference 9 is removed as per comment.
Comments:
Figure 1 - do not include acronyms (but them in brackets after the full wording
Response:
Acronyms are removed from figure 1 in lines 79-96.
Comments:
Lines 95-97: where is the verb?
Response:
Verbs are added in lines 97-101.
Comments:
Line 98 though
Response:
The word “Though” is removed in line 101.
Comments:
Line 103 why there is a question mark?
Response:
The question mark is removed in line 106.
Comments:
Acronyms are still difficult to follow, even in the text; for instance, is it really necessary to abbreviate OP for the oil price? I would strongly suggest eliminating all acronyms; this operation just takes a few minutes.
Response:
The most repeated acronyms i.e. OP, RER, BOT, and FDI are replaced by their full words in all sections of the paper.
Comments:
You may also replace GCC in the title with "Gulf" then You explain in the introduction what GCC really stands for
Response:
In the title, GCC is replaced by the Gulf Cooperation Council. The GCC is defined, at first, in lines 70-71.
Comments:
Line 209 - signified: rephrase
Response:
The sentence is rephrased in line 211.
Comments:
Line 216 - we are highlighted motivated: rephrase this "excessive" sentence
Response:
The suggested sentence is rephrased in line 218.
Comments:
229 - what is "number" of currency?
Response:
“Number of currency” is replaced by “units of currency” in line 231.
Comments:
337 - Pedroni - quote
Response:
The Pedroni reference is quoted in line 337.
Comments:
367 finding Mahmood: rephrase
Response:
The sentence is rephrased in lines 367-368.
Comments:
400 [7] asymmetrical: sentence not grammatically correct
Response:
The suggested sentence is rephrased in 403-404.
Comments:
431 delete past
Response:
The word “past” is removed in line 436.
Comments:
515-516: do not over-repeat the word "study"
Response:
The suggested sentence is rephrased to avoid the repetition of word “study” in lines 562-563.
Comments:
A real discussion is still missing ...
You may include a further figure at the end, mirroring figure 1, and explaining the results
Response:
The discussions of conclusion are extended in lines 509-521. Figure 2 is added explaining the long-run results in lines 537-559.
Comments:
Make a further effort to highlight the practical implications of your study
You may add at the end a table with a summary of the policy implications for each examined country
Response:
Table 9 is added to display a summary of the policy implications for each examined country in line 561 and reference of the table is added in lines 535-536.
Comments:
Extensive editing of English language and style required
Response:
The language and expression of the paper are improved throughout the paper.

Round 3
Reviewer 3 Report
correct line 211 importance (wrong spelling)
lines 509 ... include a short sentence explaining the impact on inflation
the rest is Ok
Author Response
Comments:
Correct line 211 importance (wrong spelling)
Response:
The spelling mistake is corrected in line 211.
Comments:
lines 509 ... include a short sentence explaining the impact on inflation
the rest is Ok
Response:
The impact of depreciation on inflation is added in lines 514-515.
